# Measurements and Accuracy Evaluation of a Strapdown Marine Gravimeter Based on Inertial Navigation

**DOI:** 10.3390/s18113902

**Published:** 2018-11-12

**Authors:** Wei Wang, Jinyao Gao, Dongming Li, Tao Zhang, Xiaowen Luo, Jinling Wang

**Affiliations:** 1Department of Earth Sciences, Zhejiang University, 38 Zheda Road, Hangzhou 310007, China; 11438025@zju.edu.cn; 2Key Laboratory of Submarine Geosciences, Second Institute of Oceanography, 36 Baochubei Road, Hangzhou 10012, China; Cdslxw@163.com; 3Beijing Institute of Aerospace Control Devices, 52 Yongding Road, Beijing 100854, China; li_dongming@163.com; 4School of Civil and Environmental Engineering, UNSW, Sydney, NSW 2052, Australia; Jinling.Wang@unsw.edu.au

**Keywords:** marine gravimetry, strapdown gravimetry system, spring-type gravimeter, gravity survey accuracy and efficiency

## Abstract

The strapdown gravimetry system uses the combination of an Inertial Measuring Unit (IMU) and a Global Navigation Satellite System (GNSS) to measure the Earth’s gravity field. Due to limited accuracies of IMU and GNSS, early strapdown gravimetry systems were more often used in airborne surveys, but less used in marine surveys. We developed a strapdown inertial navigation system (SINS), the Sea-Air Gravimeter-2Marine (SAG-2M), using novel IMU components, whose accuracy was further improved with the application of Precise Point Positioning (PPP) and enhanced algorithm, making it possible to be used in marine gravity survey. The testing results of the SAG-2M were compared to those of the Lacoste and Romberg S-129 gravimeter on the same ship in the South China Sea basin. The cruise lasted for 50 days, during which 134 effective gravity profiles were measured, resulting in 174 crossover points. The results showed that, for the SAG-2M, the root mean square (RMS) crossover points were 1.35 mGal before difference adjustment and 0.69 mGal after difference adjustment; for the S-129 gravimeter, they were 5.62 mGal and 0.95 mGal, correspondingly. In calm sea conditions, the results of the two systems were relatively consistent at all wavelengths. However, in rough sea conditions, since the SAG-2M was not affected by the cross-coupling effect, its data demonstrated less high-frequency jump. A physical platform adopted in SAG-2M can further make the transition data effective when the ship is turning around. Therefore, SAG-2M was able to improve the proportion of valid data and the efficiency of data post-processing for measurements taken during the cruise. The testing results indicate that in terms of accuracy and efficiency in the marine gravity survey, SAG-2M is better than S-129. In addition, as the miniaturization and precision of inertial components are developing continuously, SAG-2M also shows great potential in miniaturization.

## 1. Introduction

The marine gravity survey is an important tool in the fields of geodesy, geodynamics and marine science. The development of traditional marine gravimeters over nearly a hundred years has gone through three stages: pendulum-apparatus type, pendulum-rod type and axisymmetric type. Traditional scalar sea-air gravimeters have been widely used for sea-air gravity surveys, providing accuracies better than 2 mGal for spatial resolutions down to 2 km (half-wavelength) [1,2,3]. However, these systems are currently still expensive, large in size, difficult to operate, and complicated to maintain [4].

The invention of Differential Global Position System (DGPS) technology in the late 1980s improved the accuracy and continuity of positioning data, resulting in the introduction of an Inertial Navigation System (INS) composed of an Inertial Measurement Unit (IMU) and an acquisition system for airborne gravimetry. Then, in the 1990s, the Gimbaled Inertial Navigation System (GINS) and Strapdown Inertial Navigation System (SINS) were introduced [5]. The GINS directly measures gravity acceleration through a high-accuracy accelerometer on the inertial platform, thereby demonstrating the advantages of simple algorithm, stable system, mature platform, etc. A typical example of the platform airborne gravimetry system is the Russian GT-1A system, the accuracy of airborne gravity survey of which is usually around 1 mGal, and the spatial resolution of which is 2–4 km. The system is mounted on a stabilizing platform, which serves to mechanically level the sensors regardless of the carrier attitude. However, due to the limited sensitivity of modern gravity sensors, the applied correction is often insufficient. In addition, correcting the horizontal error requires the addition of a horizontal correction calculated based on the positioning data to the gravimeter measurement, which complicates not only the processing but also the evaluations of accuracy and resolution [6]. Therefore, compared with the traditional sea-air gravimeters, this type of gravimeter has not improved in reducing the size, simplifying the structure, or enhancing the data processing evaluation. Alternatively, the SINS does not employ a physical platform, but instead directly fixes the INS to the carrier, which is used in combination with the DGPS to measure the gravity value. By fixing the triaxial orthogonal accelerometer and the gyro to the carrier, this type of system can be used to measure the specific force which consists of the difference between the total acceleration and the gravitational accelerator at the measuring point inside the accelerometer. Additionally, the DGPS measures the carrier motion acceleration, which is then used to correct the motion platform acceleration. Therefore, the SINS can be used for both scalar and vector gravimetry. The Canadian Strap-down Inertial Navigation System/Differential Global Positioning System (SINS/DGPS) and the German Strap-down Airborne Gravity Meter System (SAGS) are both examples of SINS [1,7]. The SINS has multiple advantages such as small size, low cost, small power consumption, high reliability and easy operation. However, because of the system’s adoption of a “mathematical platform” to simulate the physical stabilizing platform, its requirements for the hardware temperature control technology, the inertial sensor performance and the digital signal filter are relatively high [8].

In 1995, the first airborne gravimetry flight test using a navigation-grade IMU was carried out over the Rocky Mountains by the University of Calgary. This test showed the ability of the IMU to achieve an accuracy approximately equivalent to those of the platform AIRGrav airborne gravimeter and the L and R airborne gravimeter, which was between 2 and 4 mGal, with a spatial resolution of 3 to 5 km [9]. During the same period, the Porto University developed the Litton LN-200, an airborne gravimetry system based on the relatively cheap tactical-grade IMU. The airborne survey results of the Litton LN-200 in the Azores region were first demonstrated [10] under the support of the Airborne Geoid Mapping System for Coastal Oceanography (AGMASCO) program [11]. The vertical accuracy of the Litton LN-200 system was estimated to be 5 to 10 mGal, and the spatial resolution was estimated to be 10 km [12]. In another study, Huang et al. (2012) demonstrated that their system, SGA-WZ, could obtain scalar gravity data with an accuracy better than 2 mGal and a spatial resolution of 6 km [13]. A post-processed data accuracy of better than 1 mGal was later achieved in the subsequent airborne gravity survey [14]. Diogo et al. (2015) compared the measurement accuracies of the following three strapdown inertial gravimeters: the navigation-grade IMUs, iXSea and iMAR, and a tactical-grade Litton LN-200. The results showed that, with iXSea, accuracies of 2.1 and 1.6 mGal were achieved for 1.7 and 5.0 km of spatial resolution, respectively. With iMAR, an accuracy of 4.1 to 5.5 mGal was achieved. With Litton LN-200, an accuracy of 4.5 mGal was achieved for 5 km spatial resolution [15].

At present, shipborne gravimetry is still the most important gravity measurement method in the ocean, especially in the deep-sea gravimetry. Unlike airborne gravimetry, shipborne marine gravimetry requires long-term continuous observation, which puts higher requirements on the long-term stability and the matching processing algorithm of the IMU sensors. Previously, due to technical limitations such as the drift of inertial components and the matching between DGPS and IMU, DGPS/IMU combined gravimetry systems were mostly used for airborne gravimetry, while their application in shipborne marine gravimetry was almost blank. With the development in the processing technology, the accuracy, cost control and miniaturization of the inertial components have all been significantly improved. The integration of a DGPS/IMU measuring device is considered a complement to the strapdown gravimeter, as its demonstrated long-term stability compensates for the lack of stability of the IMU in long-wavelength gravity anomaly measurements. Additionally, the application and the enhanced algorithm of Precise Point Positioning (PPP) further improves the accuracy of the strapdown gravimetry system. All of these have made it possible to use SINS in marine surveys.

This paper compared the long-term results of Sea-Air Gravimeter-2Marine (SAG-2M), a strapdown gravimeter based on DGPS and novel IMU components, with those of the traditional L and R marine gravimeter S-129 on the same ship. The post-processed data comparison and analysis results indicated that the accuracy of the SAG-2M gravimetry system was higher than S-129, and completely satisfied the current requirements of marine gravimetry. In addition, the SAG-2M gravimetry system demonstrated advantages of small size, low cost, small power consumption, high reliability and easy operation, and therefore showed broad application prospects in marine gravimetry.

## 2. Strapdown Gravimeter Measurement Principle

Based on Newton’s second law, the strapdown marine gravimeter obtains gravity acceleration and carrier attitude information through triaxial accelerometers and gyroscopes, and then realizes gravity anomaly extraction via strapdown calculation. The gravimetry mathematical model of the strapdown gravimeter is constructed based on the specific force equation of the inertial navigation system [16]:(1)$${\dot{v}}^{n}=f_{}^{n}-(2\omega_{ie}^{n}+\omega_{en}^{n})\times v_{}^{n}+g_{}^{n}$$
where $\omega_{ie}^{n}$ is the angular velocity of the Earth’s rotation, $\omega_{en}^{n}$ is the angular velocity of the rotation of the carrier coordinate system relative to the geographic coordinate system, $v^{n}$ and ${\dot{v}}^{n}$ are the carrier’s velocity and acceleration, respectively, and $g^{n}$ is the actual gravity value at the carrier’s location. The expression of the vertical gravity component can be obtained by expanding Formula (1) [9,17]:(2)$$\dot{{\dot{v}}_{U}}=f_{U}^{}+(2\omega_{ie}^{}\cos L+\frac{v_{E}^{}}{R_{N}^{}+h})v_{E}^{}+\frac{v_{N}^{2}}{R_{M}^{}+h}-g$$
where $f_{U}^{}$ is the upward specific force, L is the latitude of the carrier’s location, $R_{M}$ and $R_{N}$ denote the meridian radius and the prime vertical radius at the point on the reference ellipsoid obtained by orthogonal projection of the measuring point in the direction of the ellipsoid normal, respectively, h is the height, and $v_{E}^{}$ and $v_{N}^{}$ are the eastbound and northbound speed of the carrier, respectively. The gravity acceleration value in Formula (2) can be expressed as the sum of the normal gravity values and all disturbances, which then obtains the basic model of gravimetry:(3)$$\delta g=f_{U}^{}+(2\omega_{ie}^{}\cos L+\frac{v_{E}^{}}{R_{N}^{}+h})v_{E}^{}+\frac{v_{N}^{2}}{R_{M}^{}+h}-\dot{v_{U}^{}}-\gamma_{0}^{}$$
where $\gamma_{0}$ is the normal gravity value, and δg is the gravity disturbance.

According to the specific force equation of the inertial navigation system, the gravity anomaly calculation formula of the strapdown gravimeter is:(4)$$\delta g=f_{U}^{}+\delta a_{E}^{}-\dot{v_{U}^{}}-\gamma_{0}^{}$$
where $\delta a_{E}^{}=2\omega_{ie}^{}v_{E}^{}\cos L+\frac{v_{E}^{2}}{R_{N}^{}+h}+\frac{v_{N}^{2}}{R_{M}^{}+h}$; the accuracy of the first term $f_{U}^{}$ depends on the attitude of the inertial navigation and the measurement accuracy of the accelerometer; the third term is the corrected motion vertical acceleration. In marine gravimetry, short-period changes in the ship’s height due to ocean waves can be eliminated by digital low-pass filter processing. Consequently, the vertical motion acceleration error can be ignored. $\gamma_{0}$ should be corrected.

Therefore, the accuracy of the gravity anomaly extraction depends on the vertical component $f_{U}^{}$ of the specific force $f_{}^{n}$. We then have:(5)$$f_{}^{n}=C_{b}^{n}f_{}^{b}$$
where $f^{b}$ is the axial specific force of the carrier sensitive to the accelerometer, and $C_{b}^{n}$ is the attitude matrix of the carrier coordinate system relative to the local geographic coordinate system. Formula (5) indicates that the calculation accuracy of $f_{U}^{}$ mainly relies on the measurement accuracies of the attitude matrix of the carrier and the specific force in the carrier coordinate system during navigation.

The random error of the gyroscope and the accelerometer in the strapdown gravimeter drifts over time, which then affects the attitude matrix accuracy and the specific force measurement accuracy of the system. In order to obtain high-accuracy gravity anomaly information, it is necessary to consider the influence of inertial instrument error on the system result. Therefore, a combined navigation approach is adopted to correct for the attitude error caused by the drift of the inertial instrument. The corrected attitude matrix is then used to calculate the vertical specific force information.

## 3. Measurement Process

Between 30 October 2014 and 18 December 2014, measurements comparing the traditional spring L and R gravimeter and the SAG-2M strapdown marine gravimetry system on the same ship were carried out on the 4500-ton Xiangyanghong No. 10 Comprehensive Scientific Research Vessel (Figure 1) in the South China Sea. The survey used the Air-Sea Gravity System II marine gravimetry system produced by LaCoste and Romberg, USA. The serial number of the system was S-129 (Figure 2). The main performance indicators of the system are listed in Table 1.

The strapdown gravimeter surveyed was the Strapdown Sea-air Gravimeter SAG-2M (Figure 3) designed and manufactured by Beijing Institute of Aerospace Control Devices. The SAG-2M uses a gyroscope with a drift accuracy of 0.01°/h and an accelerometer with a measurement accuracy of 10^−6^ g. Some product parameters of the strapdown gravimeter are listed in Table 2. In order to meet the marine gravimetry requirements, the gravimeter was enhanced adaptively by carefully choosing the materials of critical components, improving the magnetic circuit design, and optimizing the process among other measures, so that the long-term stability of the parameters was improved, and the gravity sensors met the expected index requirements.

The survey lasted 50 days at an average sailing speed of 10 knots. We completed 134 effective gravity profiles with a total distance of 14,400 km (Figure 4). The two systems were placed at the same level in the same laboratory and located on both sides of the ship’s central axis. During the 50 days of gravity survey, the sea condition was generally good except for 8 days when the sea condition was harsh (level 6) due to the influence of typhoon. The sampling rates of L and R, SAG-2M and DGPS were all set at 1 Hz.

## 4. Data Processing

During the survey, both the S-129 and the SAG-2M systems continuously functioned for 50 days without any faults. After profile division, we obtained 134 profiles, resulting in 174 crossover points.

### 4.1. S-129 Gravity Data Processing Flow

The gravity data of S-129 were processed according to traditional marine gravity data processing flow, which mainly includes the following steps: (1) establish a data processing model; (2) apply zero drift correction; (3) apply draught correction; (4) apply Eotvos correction; (5) calculate normal gravity; (6) measure the absolute gravity value of the point average sea surface; and (7) calculate spatial gravity anomaly.

### 4.2. SAG-2M Data Processing Flow

The strapdown gravimeter was mounted to the measuring carrier such that the measuring axis coordinate system was parallel to the carrier coordinate system. The high-accuracy quartz accelerometer and the optical gyroscope mounted along the measuring axis were used to acquire the upward specific force (the sum of conservative and non-conservative forces) information and the angular motion information of the three axes of the carrier. The GNSS PPP/INS navigation system [18] was adopted to provide a high-accuracy mathematical stabilizing platform to the gravity sensors. The vertical component of the specific force information was obtained by calculating the attitude projection, which was then subtracted by the vertical motion acceleration estimated by PPP (or DGPS) to calculate the difference. Following that, the gravity anomaly extraction on the track was achieved by the combining filter and the correction algorithm, and finally the scalar gravity information on the surveyed track was obtained.

The strapdown gravimeter data processing flow is divided into the following 5 steps, as shown in Figure 5:Calculate the attitude, speed and position information of the carrier at the current moment based on the angular velocity and the linear acceleration of the carrier at the current moment measured by the inertial instrument;Perform combined navigation based on the carrier speed and position information provided by PPP (or DGPS) to correct for the inertial attitude error;Use the corrected attitude matrix to project the specific force information to the local geographic coordinate system, so as to obtain the corrected vertical component of the specific force;Calculate the Eotvos correction item and the normal gravity value of the carrier based on the speed and position information provided by PPP (or DGPS);Design a low-pass filter based on the noise characteristics to eliminate the measurement noise, so as to obtain the gravity anomaly information.

The inertial system error state equation used in the Kalman filter in Figure 5 can be expressed as [19]:(6)$$\dot{X}=FX+GW$$

Considering the amount of calculation performed by the filter, the state vectors used can be simplified as appropriate. Using the system state vector $X={\left(\delta L\;\delta \lambda \;\delta v_{E}^{}\;\delta v_{N}^{}\;\varphi_{E}^{}\;\varphi_{N}^{}\;\varphi_{U}^{}\;\nabla_{ax}^{}\;\nabla_{ay}^{}\;\nabla_{az}^{}\;\varepsilon_{gx}^{}\;\varepsilon_{gy}^{}\;\varepsilon_{gz}^{}\right)}_{}^{T}$ as an example, where $\varphi_{E}^{}$, $\varphi_{N}^{}$ and $\varphi_{U}^{}$ are the eastbound, the northbound and the upward attitude error angles, respectively, and $\nabla_{ax}^{}$, $\nabla_{ay}^{}$, $\nabla_{az}^{}$, $\varepsilon_{gx}^{}$, $\varepsilon_{gy}^{}$, and $\varepsilon_{gz}^{}$ are the drift errors of the accelerometer and the gyro in the *x*, *y* and *z* direction, respectively. F is a 13×13 square matrix, G is a 13×6 matrix, and W is the noise matrix. The system observation formula is:(7)Z=HX+V

If we take the difference between the speed derived from the inertial instrument and the speed calculated by PPP (or DGPS) as the observation value, then $Z={\left(v_{INS}^{E}-v_{GPS}^{E}\;v_{INS}^{N}-v_{GPS}^{N}\right)}_{}^{T}$; V is the observation white noise. In the observation matrix H, except for H(1,3) and H(2,4) whose values are 1, the remaining elements have a value of 0.

Unlike traditional spring sea-air gravimeters, the strapdown gravimeter directly measures the specific force information of the carrier system and the angular motion information of the carrier. Therefore, during measurement, the requirements on the motion condition of the carrier are less strict. In addition, when processing gravity data, the strapdown gravimeter does not have to go through the time- and energy-consuming profile division step like the traditional spring gravimeters do. Therefore, it not only improves efficiency but also increases the ratio of valid measurement data, as the data measured when the carrier is turning are also effective.

## 5. Survey Data Comparison

According to the free-air anomalies (FAA) calculation Formula (4) and formula $\Delta g_{F}^{}=g-\gamma_{0}^{}$, g is absolute gravity, $\gamma_{0}^{}$ is normal gravity. We calculated the FAA of SAG-2M and S-129. Figure 6a,b compare the data collected under normal conditions. The two systems demonstrate a consistent general trend and have their own advantages at the specific resolution. With the same filtering, S-129 shows a better 1 km resolution than SAG-2M, but SAG-2M shows a better 1.5 km resolution. Both the SAG-2M and the S-129 systems can achieve a resolution higher than 1 km and an accuracy higher than 1 mGal with 180 s filtering.

Figure 6c,d show data collected on 18 November and 7 December. It is noted from the figures that the data quality of S-129 is far worse than that of SAG-2M. In Figure 7c S-129 demonstrates a strong noise with a period of 6 min, while in Figure 6d S-129 demonstrates a strong high-frequency noise. Alternatively, the data acquired by SAG-2M are stable and show a smooth trend, the quality of which are not too different from that displayed in Figure 6a,b. This is because the sea condition was relatively poor in these two measurement periods. The S-129 platform is easily influenced by the motion state of the carrier, resulting in a strong cross-coupling effect that affects the data acquisition accuracy, whereas the SAG-2M adopts a physical platform and therefore presents no cross-coupling effect. Consequently, compared to traditional platform gravimeters, SAG-2M is more adaptable to the environment.

To obtain an overall evaluation of the S-129 and the SAG-2M data, the average difference and the standard deviation of the profiles of S-129 and SAG-2M before and after difference adjustment, as well as the average difference and the standard deviations of the differences between the gravity anomaly data of S-129 and SAG-2M, are calculated. Before adjustment, the average difference between the overall FAA of S-129 and SAG-2M is 7.5 mGal, and the standard deviation is 1.5 mGal; the average difference displays a linear increase trend, as shown in Figure 7a. A certain linear relationship between the average difference and the standard deviation is observed, as shown in Figure 8a. After difference adjustment, the average difference is 9.2 mGal, and the standard deviation is 1.4 mGal. No correlation is observed between the average difference and the standard deviation, as shown in Figure 8b. The results indicate that the linear correlation is caused by a semi-systematic error.

Comparison analysis of the raw data shows that this semi-systematic error is caused by gravimeter drift. The drift of the S-129 spring gravimeter during the entire cruise is 4.94 mGal. However, the SAG-2M gravity survey drift is a random error that exhibits a strong linear correlation, which can be decreased during long-term measurement. Therefore, it is concluded that the cause of the phenomenon observed in Figure 7a and Figure 8a is the drift of the two gravimeters, and that the impact of S-129 drift is larger than that of SAG-2M.

In the average difference statistics chart, the difference between the two systems is larger during the periods of 16 November to 23 November, and1 December to 8 December, whose corresponding sea condition is relatively poor. During these periods, the data quality of S-129 is affected by the sea condition and has a higher fluctuation, resulting in a large difference between the two systems. This is also reflected in the profile comparisons shown in Figure 6c,d.

Figure 9 shows the gravity anomaly maps of the survey area acquired by SAG-2M and S-129. The source data of Figure 9a is all the data collected by the SAG-2M gravimeter in the survey area, including those acquired when the ship is turning or parked. Figure 9b,c show the resultant gravity anomaly maps of SAG-2M and S-129, respectively, post traditional spring marine gravimeter profile division processing, with profiles shown in Figure 2.

All three sets of data are applied with the same interpolation method, and are not fine-filtered or smoothed. By comparing numerical and spectral analysis, in Figure 9b,c, the overall resolution of the data in Figure 9b is higher than that of the S-129; particularly in areas near the island reef where the gravity value changes sharply, Figure 9b contains more details. In the box outlined in Figure 9c, where the profile corresponds to the data shown in Figure 6c, because the sea condition is poor, cross coupling has a larger influence on the data quality compared to the corresponding area in Figure 9b. Therefore, difference in the data quality can be clearly observed. When comparing Figure 9a,b, Figure 9a shows a higher resolution, especially in the northern part of the survey area. This is because the adoption of a physical platform by SAG-2M makes it unnecessary to perform profile division in data processing. Therefore, the gravity data acquired when the carrier is turning are also effective. Comparing Figure 9a with the profiles in Figure 2, it is noted that SAG-2M acquires more effective data than S-129. Consequently, SAG-2M has a unique advantage in data processing and acquisition, and can greatly improve the efficiency of gravity surveys in the future.

The data of L and R S-129 sea-air gravimeter has 174 crossover points, whose average difference and root mean square (RMS) are 4.94 mGal and 5.62 mGal before difference adjustment, respectively, 1.34 mGal and 0.95 mGal after difference adjustment. The data of SAG-2M’s average difference and RMS are 1.94 mGal and 1.35 mGal before difference adjustment, respectively, 0.74 mGal and 0.69 mGal after difference adjustment.

The comparison results of the crossover points of the S-129 and the SAG-2M gravimetry system are shown in Figure 10. The crossover-point errors of both systems are normally distributed, but the crossover-point error of SAG-2M is more concentrated, indicating that the SAG-2M system has a higher overall measurement accuracy than the S-129 marine gravimeter.

## 6. Conclusions and Outlook

In this paper, by processing and analyzing the gravity data acquired by the novel strapdown gravimeter SAG-2M and the traditional spring gravimeter S-129 on the same ship, the performance of the SAG-2M gravimeter under the same working conditions was evaluated.

The SAG-2M adopted new IMU components and combined PPP (DGPS) data to improve the calculation algorithm, thereby solving the problems of long-term instability of IMU components and poor long-wavelength accuracy. In terms of data processing, the SAG-2M directly enhanced the accuracy in calculating the vertical acceleration of the carrier through introducing PPP to improving the accuracy of navigation data, thereby leading to an increased gravity survey accuracy. Compared to traditional spring gravimeters, the strapdown gravimeter SAG-2M has the following advantages:The system has a smaller weight and size, a lower power consumption and a simpler structure, and is therefore more adaptable to the working environment;The system has a higher accuracy and stability, and presents a stable linear drift that can be effectively eliminated by digital correction;The acquired data are more accurate, the processing is simpler, and the measurement is more efficient; andThe average difference and the standard deviation of the gravity data acquired by the SAG-2M strapdown inertial gravimeter after difference adjustment are 0.74 mGal and 0.69 mGal, respectively, which are better than those of 1.34 mGal and 0.95 mGal of the S-129 gravimeter.

Therefore, the strapdown gravimeter SAG-2M with its many advantages can fully satisfy the requirements of marine gravity survey and as such provides an alternative option to it.

Besides the above advantages, the strapdown gravimeter also presents great potential. With further human exploration of the ocean, the research scale will be more refined and the data quality requirements will be higher. Therefore, the marine gravity survey will be developed towards deep sea and near bottom surveys, which is more demanding on the size and power consumption of marine gravimeters. After decades of development, traditional gravimeters have limited potential in improvement in miniaturization, and therefore cannot satisfy the size and power consumption requirements of future deep-diving devices and autonomous underwater vehicles (AUVs). Alternatively, free from the traditional gravimetry platform, strapdown inertial gravimeters such as SAG-2M are simpler in structure, and the inertial components have more substantial potential in miniaturization and reducing energy consumption when compared to the spring components. 

## Figures and Tables

**Figure 1 sensors-18-03902-f001:**
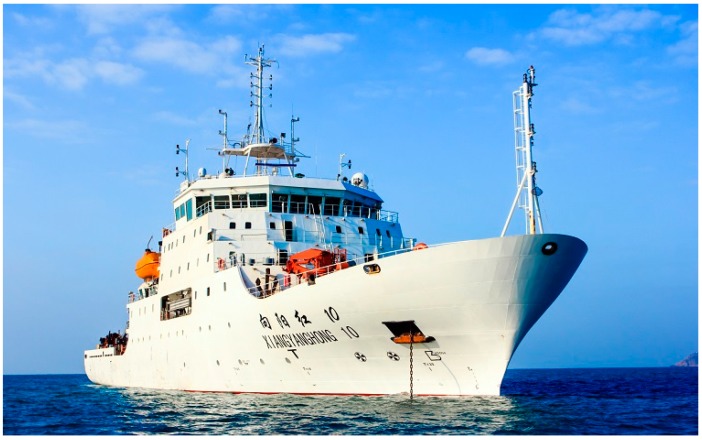
Survey carrier: Xiangyanghong No. 10 Vessel.

**Figure 2 sensors-18-03902-f002:**
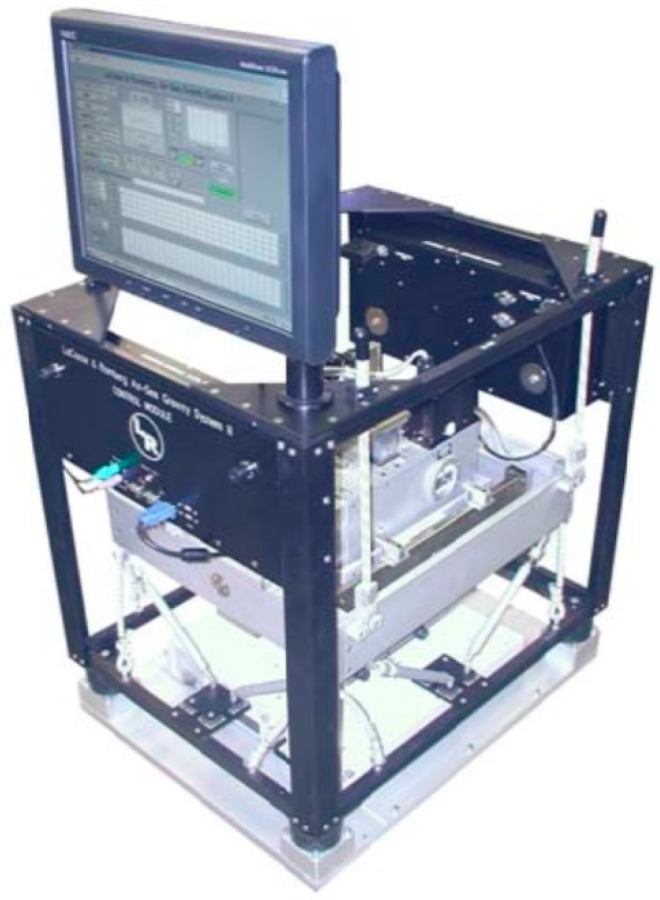
LaCoste and Romberg S-129 gravimeter.

**Figure 3 sensors-18-03902-f003:**
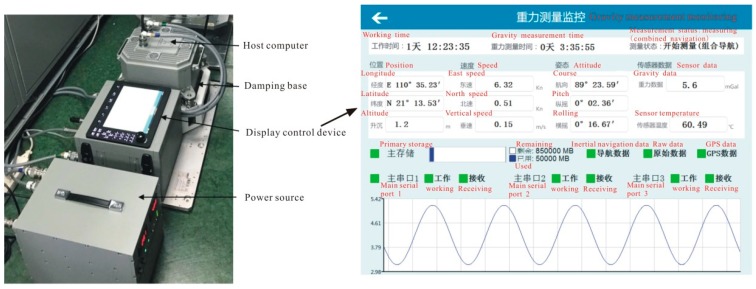
The Sea-Air Gravimeter-2Marine (SAG-2M) system and its operating interface.

**Figure 4 sensors-18-03902-f004:**
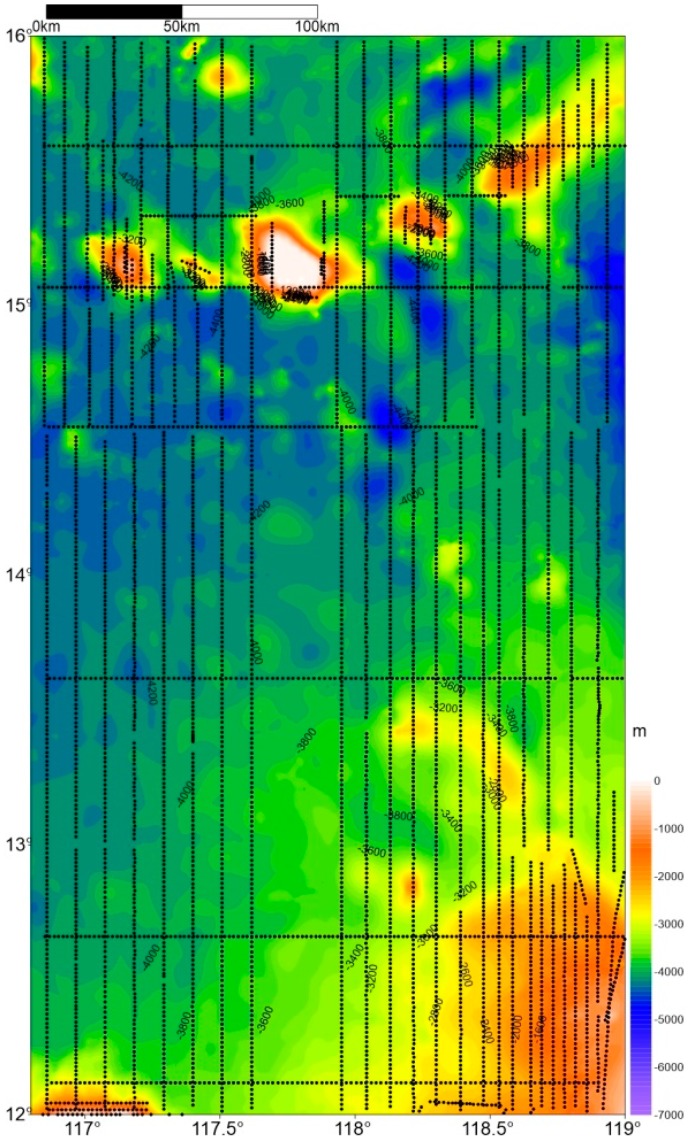
Survey area profile plot.

**Figure 5 sensors-18-03902-f005:**
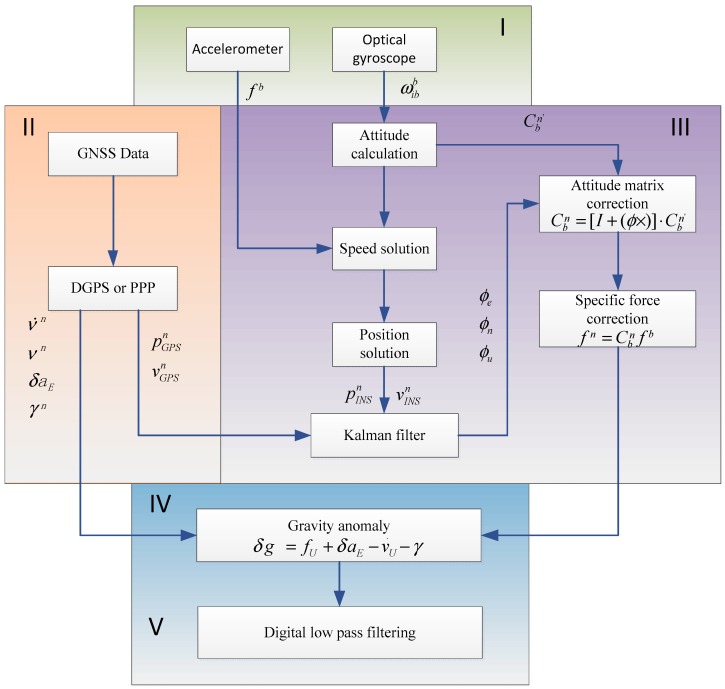
Strapdown gravimeter data processing flow.

**Figure 6 sensors-18-03902-f006:**
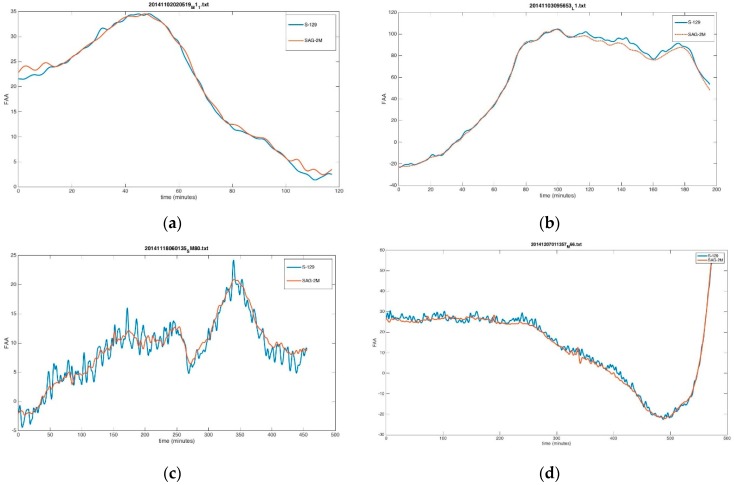
Profile comparison (blue profiles are the data acquired by S-129, and red profiles are the data acquired by SAG-2M).

**Figure 7 sensors-18-03902-f007:**
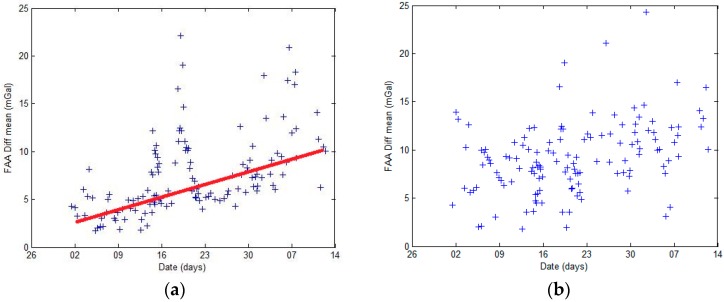
(**a**) Profile average difference before difference adjustment; (**b**) Profile average difference after difference adjustment.

**Figure 8 sensors-18-03902-f008:**
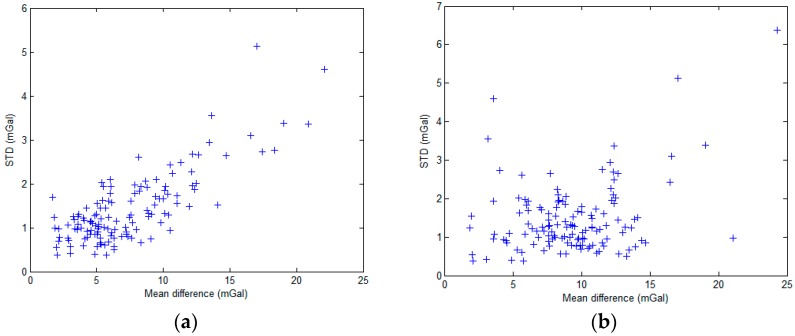
(**a**) Correlation between average difference and standard deviation before difference adjustment; (**b**) Correlation between average difference and standard deviation after difference adjustment.

**Figure 9 sensors-18-03902-f009:**
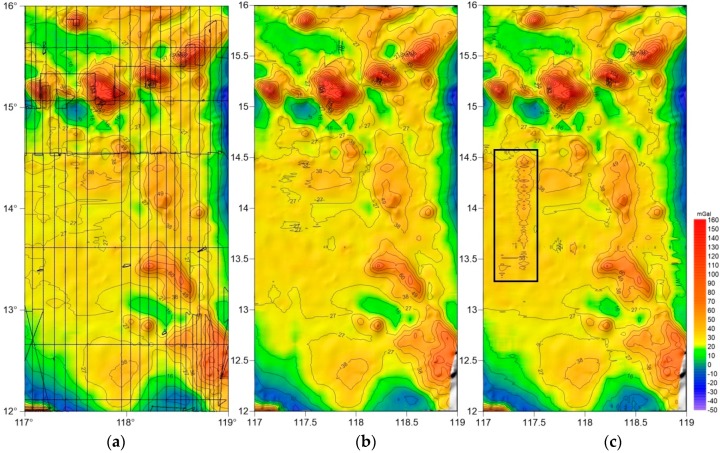
(**a**) SAG-2M gravity anomaly map before profile division; (**b**) SAG-2M profile data gravity anomaly map; (**c**) S-129 gravity anomaly map.

**Figure 10 sensors-18-03902-f010:**
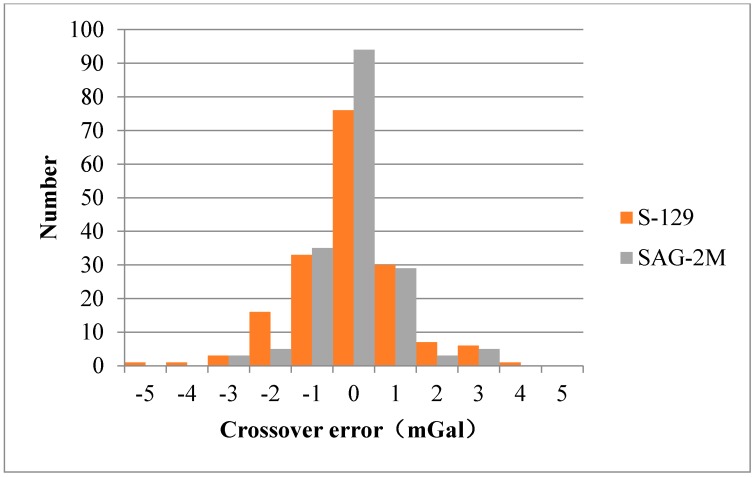
Comparison results of the crossover points of the S-129 and the SAG-2M sea-air gravimeters.

**Table 1 sensors-18-03902-t001:** Basic parameters of the L and R S-129 gravimetry system.

Item	Parameter
Sensor type	Zero-length spring/pendulum
Measurement principle	Pendulum movement rate
Sensor accuracy	10 µGal
System resolution	0.01 mGal
System zero drift	<3 mGal/month
System constant temperature	Factory rated ± 0.01 °C
Platform control	21-bit DSP computer numerical control
Dimensions	71 cm × 56 cm × 84 cm
Weight	116 kg

**Table 2 sensors-18-03902-t002:** Basic parameters of the strapdown gravimetry system.

Item	Parameter
Total weight	60 kg
Volume	660 × 700 × 290 mm
Operating voltage	28 V ± 4 V
Range	20,000 mGal
Maximum inclination (roll or pitch)	±45°
Working environment temperature	0 to 45 °C
Gyro accuracy	0.01°/h
Accelerometer accuracy	10^−6^ g
Static measurement accuracy	<0.4 mGal
Offshore measurement accuracy	<1.0 mGal

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
