# Peer review of "Measurements and Accuracy Evaluation of a Strapdown Marine Gravimeter Based on Inertial Navigation"

_sensors, 2018, doi:10.3390/s18113902_

Round 1
Reviewer 1 Report
Comments to the authors
This paper presents an interesting comparison of gravity surveying methods carried out at sea, using two kinds of gravity meter based on completely different measurement principles. On the one hand, the LaCoste & Romberg S-129 air-sea gravity meter consists of a spring-type relative gravity meter mounted on a mechanical gyro-stabilized platform. On the other hand, the SAG-2M marine inertial gravity meter is based on an Inertial Measurement Unit (IMU) securely attached onboard, consisting itself by both one 3-axis accelerometer and one 3-axis gyro. Such system provides the specific force, the attitude orientation angles (yaw, pitch, roll) and position vector coordinates of the carrier vehicle with a high sampling frequency, that can be then combined with GNSS-derived position vector coordinates in order to compute the gravity vector at sensor's location. Thanks to the development of silicon micro-machined inertial sensors, the common size of IMUs has dramatically decreased, thus allowing the manufacture of small-size strapped down inertial gravity meter to be contemplated. Although the SAG-2M cannot be counted as a small -size gravity meter (weight: 60 kg, volume: 660 x 700 x 280 mm3), it can be a step towards developing small-size, low-cost strapped down gravity meters, suitable for the gravity surveys carried out onboard light carrier vehicles, even unmanned, operating at sea, under sea, and in the air.
The comparison is based on the data acquired on 134 gravity profiles at sea, the overall distance of which is at 14,400 km. Given an average sailing speed of 10 knots (18,52 km/h) and a sampling frequency at 1 Hz, the mean distance between two measurements along each profile is equal to 0.005 km, thus giving a total of 2,880,000 measurement points. Such a great volume of data allows a comprehensive statistical analysis to be conducted. In my opinion, that is a great strength of the work that should have been even more highlighted in the paper.
On the whole, the paper is correctly written (only a few misspelling and spelling mistakes) and the cited references are abundant and relevant. The theory, gravity survey methodology and data processing outlined in the paper (sections 2, 3, 4) are generally clearly explained. The examples of free-air gravity anomalies measured along marine profiles testifying to the SAG-2M's performance are rather convincing (section 5). However, I would like to make some suggestions in order to clarify and/or to complete some aspects of the paper described subsequently.
General comments and suggestions
In the section “Survey data comparison” (section 5), the free-air anomalies (FAA) along four marine profiles are shown graphically in figures 6(a) - (d). However, there is no equation for defining what is called free-air anomaly in the paper. If the free-air anomalies that are actually plotted in figures 6(a)-(d) are those given by equation 4 which defines the gravity anomaly δg, it would be desirable to indicate it at the beginning of section 5.
In the same section, you mention the respective spatial resolutions of the free-air anomalies calculated from the data acquired on the same profiles by means of S-129 and SAG-2M gravity meters. At that stage, the way to estimate the spatial resolutions of the free-air anomalies has to be clearly explained. Indeed the values given in the paper cannot be efficiently attested by a simple review of free-air anomaly variations. You have to convince your reader of the reliability of your spatial resolution estimates, why not by a spectral analysis. The comment is applicable to the gravity anomaly maps showed in figures 9 (b) and 9 (c), for which you consider that the spatial resolution of the gravity anomaly map deduced from SAG-2M's gravity data (Fig. 9 (b)) “is higher than” that deduced from S-129's gravity data.
Figure 8 (a) in section 5 indicates that, to a certain extent, there exists a linear relationship between the average differences of the gravity values provided by the S-129 and the SAG-2M gravity meters respectively before difference adjustment, and the corresponding standard deviation. According to you rationale, this finding suggests that the gravity measurements are affected by a random drift among other effects, that touches exclusively the SAG-2M's gravity data. I think that you could comment in more detail on this delicate issue and at least offer one compelling argument.
My last comment relates to the GNSS data processing. As shown in figure 5, GNSS data are involved in the SAG-2M's gravity data processing flow (see box II), where they contribute to both the calculation of the attitude matrix correction after Kalman filtering (box III) and the computation of the gravity anomaly (box IV). As you rightly mentioned in the paper, the processing of GNSS data can be performed by DGPS or PPP method. With an eye to comparing the accuracy and the efficiency of the two methods, I suggest you directly compare the gravity anomalies determined respectively by a processing using either DGPS method or PPP method. The paper would therefore be of greater merit.
Finally, some additional comments are directly written down in the printed version of the paper in attachment.

Author Response
Thank you very much for your valuable suggestions. Your suggestion is very important for the article to improve quality . At the same time, these suggestions have benefited me a lot. Thanks again.
Reviewer 2 Report
Over all it is a good paper that deserves to be published here. The developments are clear. The results are very good. There are only a few minor questions in the attached comment boxes. It would be wonderful if the authors can fix them. I would like to see the final version before it gets published.
Author Response
Thank you very much for your valuable suggestions. Your advice has benefited me a lot. Thanks again. Modified article see attachment.
Reviewer 3 Report
The paper shows the results of the use of a strap down gravimeter for marine gravity measurements. A new system has been developed and it has been tested over a significantly large campaign. The processing method is sound, the results have been clearly presented and they are of great interest with respect to the measurement of the gravity field.Please correct gravitational field to gravity field.
Author Response
Thank you very much for your valuable suggestions. Your suggestion is very important for the article to improve quality. Thanks again.
The gravitational field has been corrected to the gravity field.
Round 2
Reviewer 2 Report
Now. It is good enough. I have no further questions.